# Quantifying nursing care delivered in Kenyan newborn units: protocol for a cross-sectional direct observational study

David Gathara,[1] George Serem,[1] Georgina A V Murphy,[1,2] Nancy Abuya,[1,3] Rose Kuria,[4] Edna Tallam,[5] Mike English[1,2]

[1]Department of Public Health Research, Kenya Medical Research Institute/Wellcome Trust Research Programme, Nairobi, Kenya
[2]Nuffield Department of Medicine and Department of Paediatrics, University of Oxford, Oxford, UK
[3]Department of Curative and Preventative Services, Nairobi City County, Nairobi, Kenya
[4]Department of Nursing, Kenya Medical Training College, Nairobi, Kenya
[5]Department of Registration and Licensing, Nursing Council of Kenya, Nairobi, Kenya

**Correspondence to**
Dr David Gathara;
DGathara@kemri-wellcome.org

## ABSTRACT

**Introduction** In many African countries, including Kenya, a major barrier to achieving child survival goals is the slow decline in neonatal mortality that now represents 45% of the under-5 mortality. In newborn care, nurses are the primary caregivers in newborn settings and are essential in the delivery of safe and effective care. However, due to high patient workloads and limited resources, nurses may often consciously or unconsciously prioritise the care they provide resulting in some tasks being left undone or partially done (missed care). Missed care has been associated with poor patient outcomes in high-income countries. However, missed care, examined by direct observation, has not previously been the subject of research in low/middle-income countries.

**Methods and analysis** The aim of this study is to quantify essential neonatal nursing care provided to newborns within newborn units. We will undertake a cross-sectional study using direct observational methods within newborn units in six health facilities in Nairobi City County across the public, private-for-profit and private-not-for-profit sectors. A total of 216 newborns will be observed between 1 September 2017 and 30 May 2018. Stratified random sampling will be used to select random 12-hour observation periods while purposive sampling will be used to identify newborns for direct observation. We will report the overall prevalence of care left undone, the common tasks that are left undone and describe any sharing of tasks with people not formally qualified to provide care.

**Ethics and dissemination** Ethical approval for this study has been granted by the Kenya Medical Research Institute Scientific and Ethics Review Unit. Written informed consent will be sought from mothers and nurses. Findings from this work will be shared with the participating hospitals, an expert advisory group that comprises members involved in policy-making and more widely to the international community through conferences and peer-reviewed journals.

## Strengths and limitations of this study

► The use of direct observational methods to quantifying nursing care delivered or left undone is an approach that has not been previously used in low/middle-income countries.
► Different sectors (public, private for profit, private not for profit) in health service provision have been included in this study.
► The study provides a 24-hour assessment of neonatal nursing care provision including care provided on weekends and weekdays.
► Our scope is limited by the few number of newborns to be observed within each hospital.
► Despite our efforts to minimise the Hawthorne effect, we cannot rule out the possibility of nurses changing the way they provided care during the observational periods.

only slow declines in neonatal mortality. As a consequence, about 45% of mortality for children under 5 years is attributable to neonatal mortality.[1] Of these neonatal deaths, approximately 75% occur in the first 7 days of life and half of these within 24 hours of life.[2 3] A recent review by Bhutta *et al* indicated that high-impact, low-cost interventions could avert more than 71% of neonatal deaths with 82% of this effect being attributable to facility-based care.[4] However, reports from low-income settings highlight that the quality of newborn care in health facilities is often poor.[5–7] Therefore, strengthening the quality of facility-based care for newborns will be essential in improving newborn outcomes.

Human resources for health inadequacies is a major factor limiting delivery of quality neonatal services.[8] Globally, the shortage of health workers is estimated currently at over 7 million and by 2035 the deficit is estimated to be 12.9 million.[9] The shortages in the available workforce are worst in low/middle-income countries (LMICs) where

## INTRODUCTION

Despite progress globally, most African countries including Kenya have made insufficient progress in reducing child mortality. In most countries, this can be partly attributed to

inequitable distribution of available health workers may compound the problems. In Kenya, Wakaba *et al* reported that public sector nursing densities ranged between 0.008 and 1.2 per 1000 population across counties[10] compared with an internationally suggested minimum health workforce threshold of 2.5/1000 population for doctors, nurses and midwives. In Nairobi County, the nurse densities ranged between 0.21 and 0.40 per 1000 population.[10]

There is little specific exploration of the impact of nursing workforce shortfalls on inpatient care in LMICs. Yet to improve quality of care, it is essential that we understand who delivers care (what tasks are done by whom), how care is delivered (how are tasks performed) and critically analyse what tasks are left undone. In most health systems, nurses are gatekeepers of the healthcare being delivered. They are vested with the responsibility of delivering interventions prescribed by other providers (doctors, nutritionists, etc) in addition to providing nurse-initiated interventions.[11] As a consequence, few interventions reach the patient without the involvement of the nurse. Yet the few existing evaluations of the quality of care provided to newborns in LMIC have focused on the more medical aspects of care.[6 12–14] In LMIC facilities, large patient workloads, insufficient staff and resources, urgent patient situations and unexpected rise in patient volume and/or acuity on the unit (among other factors) might result in all facets of nursing care being delayed or neglected. This phenomenon has been described as 'implicit rationing',[15] 'missed care'[16] or 'unmet nursing care needs',[17] 'care left undone',[18] or 'task incompletion'.[19] Hereafter, we use the term missed care to encompass all of these terms. Such missed care may have a particularly devastating impact on outcomes in newborn units where nurses are the primary caregivers to this highly dependent group.

Justifying a focus on missed nursing care several studies have reported associations between missed care and patient outcomes.[20–22] Although there is growing literature on missed care the majority is from high-resource settings, with only one study in South Africa[23] providing an early formal attempt to quantify the extent of the problem in this middle-income country. Furthermore, almost all the literature on missed care is based on nurse surveys with only two focusing on newborn care provision.[24 25] Although, nurse surveys on missed care have proven useful, there is a call to undertake more research with a special focus on objective observational methods as no studies of this type were identified in a recent systematic review.[22] The proposed study aims to characterise the care delivered (tasks done or left undone and who does these tasks) to newborns receiving care within newborn units in Nairobi, Kenya by making direct observations of care being provided. This will provide in-depth and objective insights on missed care with particular reference to locally agreed standards for nurses providing neonatal care.

## METHODS AND ANALYSIS

This is a cross-sectional study that will involve direct observation of care provided to individual newborns with the aim of describing the essential neonatal nursing care given or missed within newborn units. It will be undertaken in six hospitals in Nairobi City County, Kenya, in the period from 1 September 2017 to 30 May 2018.

### Study site

The proposed research work will be undertaken as part of a broader set of work being conducted in collaboration with Nairobi City County. This collaboration includes work to characterise all the facilities providing inpatient newborn care 24 hours, 7 days a week (hereafter referred to as 24/7) in Nairobi,[26] quality of clinical care provided to newborns[27] and ethnographical work to understand the wider context and practice of neonatal nursing. Based on findings from the broader study, Nairobi County has 34 health facilities providing 24/7 inpatient newborn care, of these, two small health facilities declined to take part in prior quality of care surveys and were estimated to have less than 50 neonatal admissions each per year.[27] Excluding the military health facility with restrictive admission policy, the remaining 31 health facilities that form the population for this study provide 99% of all inpatient neonatal care.

This current study will focus on primary referral (secondary care) facilities that have more than 100 neonatal admissions annually. As such, the two health facilities that declined to participate in prior quality of care surveys do not meet this criterion and are unlikely to introduce selection bias. Thirteen facilities meet these criteria and together provide care to over 96% of the sick newborn population accessing care within Nairobi County.[27] These 13 facilities will be stratified by workload (newborn admissions per year ≤500 low; >500 high) and six health facilities purposefully selected to ensure representation of two hospitals in each of the public, private-not-for-profit and private-for-profit sectors, with one high and one low workload facility in each sector. Purposeful selection will be used as it is important in this sensitive and innovative work to have the strong support of the hospital administration. This initial work will therefore help illustrate the nature and magnitude of the challenge of missed care but not make claims to provide a statistically representative picture of missed care which would be challenging given the great diversity of facilities found in Nairobi, with some health facilities having as low as 10 neonatal admissions per year.[27]

### Study populations

All newborns admitted within the newborn unit in the six selected health facilities over the period of the study in each facility will form the potential study population. However, newborns meeting the following exclusion criteria will not be observed: (1) newborns requiring specialised treatment to whom the draft minimum nursing standards for neonatal care[28] may not be

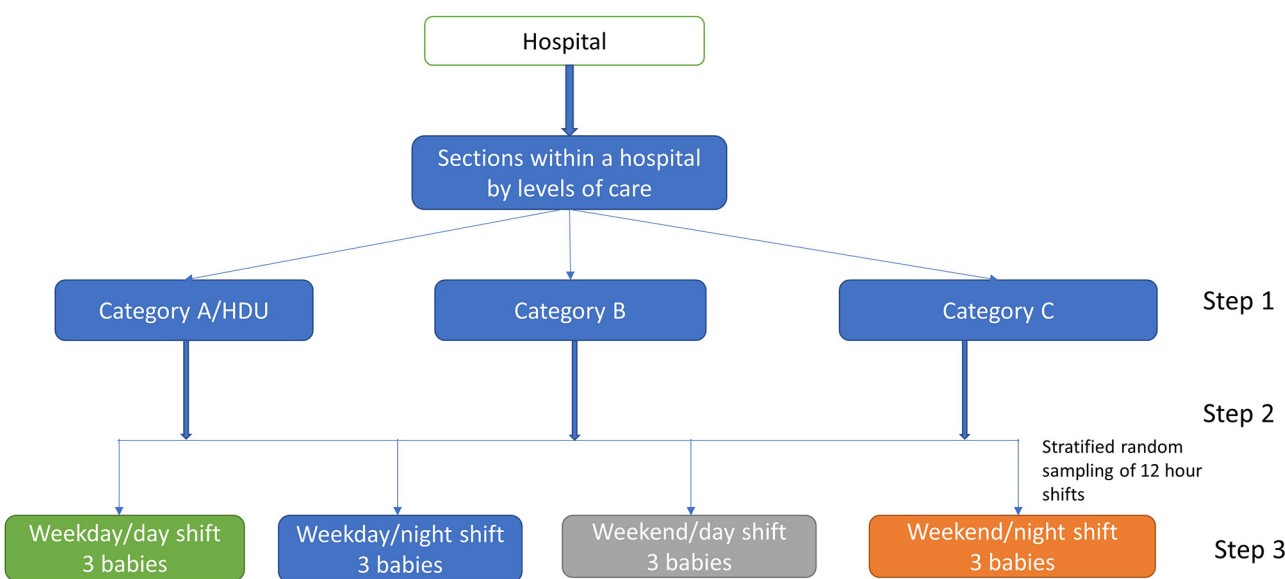

**Figure 1** Steps for the sampling procedure. Multistage sampling procedure within a hospital for selecting newborns for direct observation. HDU, high dependency unit.

applicable, for instance, newborns with gross malformations or those receiving post-operative care, (2) newborns who are critically ill and at risk of death within a 12-hour observation period as defined by the clinician in charge of the newborn unit for whom observation might cause distress to families, (3) newborns for whom guardians do not provide consent and (4) newborns receiving care from nurses or guardians who decline to be observed in the care provision process.

## Sampling procedures

To describe the care being provided to newborns admitted in newborn units and the spectrum of inpatient services they receive, we are aiming to sample time in 12-hour shifts randomly using the steps described in figure 1.

### Step 1

Care within the newborn unit can be organised in a way that babies requiring different levels of care are in different sections/rooms. Intensive care employing invasive mechanical ventilation is not available outside tertiary hospitals in Kenya and thus definitions for different levels of newborn care provided by county hospitals were adopted. These are taken from draft nursing standards for neonatal care that categorise newborn unit sections (and thus babies) in the following way.[28] Category A (high dependency unit): Babies on oxygen/continous positive airway pressure and intravenous fluids who are often acutely ill and unstable and require the closest monitoring. Category B: Babies who have stabilised but may still be ill and receiving, for example, assisted feeding (nasogastric feeds), intravenous drugs or being observed for convulsions or apnoea. Category C: Babies who are quite stable who should be receiving kangaroo mother care or stable abandoned babies, or recovering babies requiring completion of treatment such as the last doses of antibiotics or transitioning to oral feeding. Our

primary sampling strategy will be based on identifying these organisational subsections (category A, B, C rooms or incubator/cot spaces) in each facility. Where facilities have no clear organisational demarcation into category A, B and C subsections, we will adapt the observations to suit the organisation of care in each facility. However, we will endeavour to identify and select, for observation, newborns with varying degrees of illness severity who would be classified as meeting criteria for category A, B or C in such settings.

### Step 2

In each hospital, stratified random sampling will be used to generate a random sample of 12 shifts/time blocks of 12 hours (144 observation hours per hospital) from within a 3-week period stratified by disease severity (category A, B, C), weekdays and weekends as well as night and day shifts as care has been shown to vary across these periods.[29 30] The 12 shifts will be divided equally across the newborn unit sections within a hospital where there is more than one section. Where all babies are cared for in one room without clear sub-sections, all observation periods will be conducted in this same setting with efforts to observe babies requiring different levels of care.

### Step 3

We feel that it is logistically feasible for one observer to make direct observations of three babies located in adjacent cots in the same ward area at one time. Therefore, for each shift and section, three babies who meet the inclusion criteria will be purposefully selected to ensure babies are in one ward area and within close proximity to allow direct observations. In smaller units where the number of babies per section may be less than three, we will observe all the babies available in the section and classify them as eligible for care in category A, B or C.

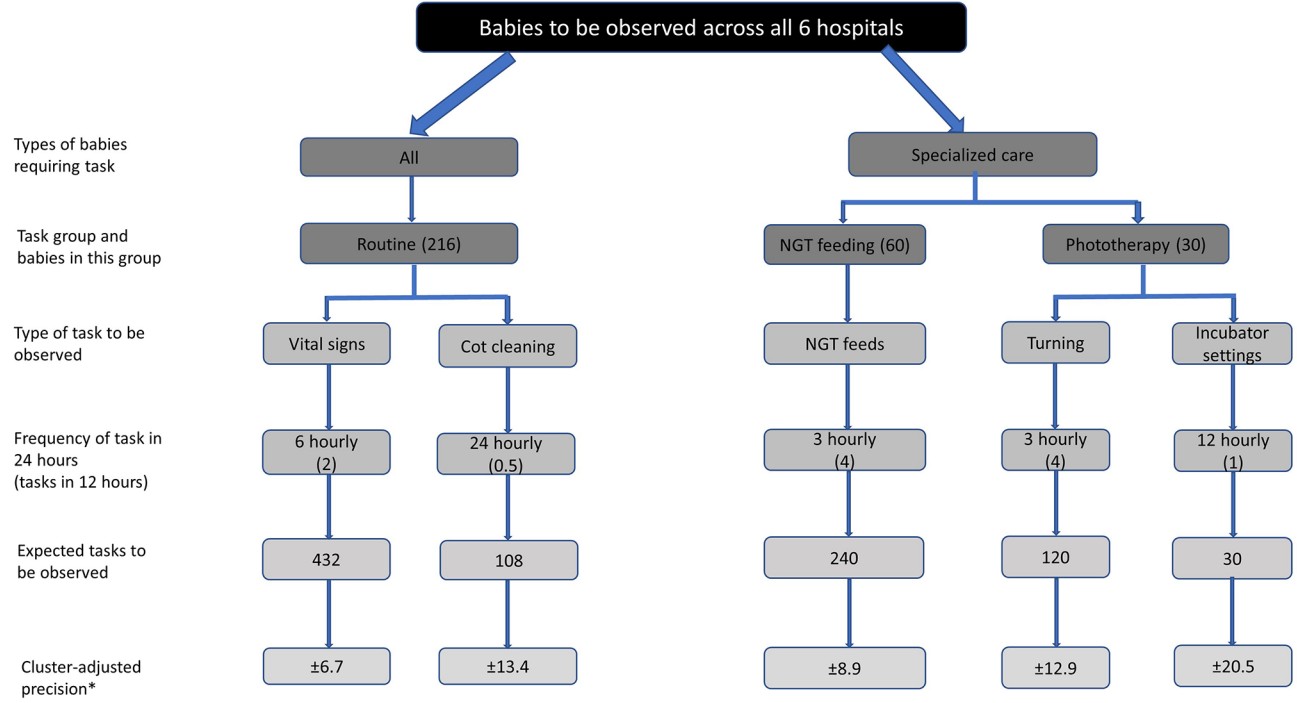

**Figure 2** Precision levels for different newborn subpopulations and tasks estimated levels of precision for the different newborn subpopulations and tasks observed that the study will report since not all newborns observed will require all tasks.

## Sample size

This is an exploratory cross-sectional study and as such, we illustrate the precision with which we can report proportions of tasks done (or not done) assuming different-sized denominators. The size of the denominator is related to the recommended frequency that tasks should be performed (see figure 2). To estimate the precision of reporting, we have used a sample estimation approach for cluster designs and assuming a design effect of 2 to adjust for clustering of observed tasks around individual newborns within hospitals.

At most we expect to recruit 216 newborns across 6 hospitals (36 newborns per hospital with 3 babies for each of the 12 shifts/time blocks). For tasks that would be conducted with an expected frequency of once in 12 hours (eg, intravenous penicillin administration) we might therefore observe 216 task opportunities. Taking a (statistically) conservative assumption that on 50% of expected occasions the task is observed to be done then we could report the proportion of such a task being done with a precision of ±13.4%. Similarly, if a task should be conducted every 6 hours then the denominator would be 432 expected tasks and if half were observed the precision of this estimate would be ±6.7%.

Not all babies observed will require all tasks. For example, some babies may not be receiving intravenous drugs. This will reduce the effective size of the denominator and reduce the precision we can report. Similarly, if babies have to leave the area of observation (eg, they are moved to a new ward area or are sent for X-ray etc), this may reduce the number of expected tasks that can

be observed. In addition, failure to recruit three babies at each 12-hour shift at each facility may also reduce the number of tasks observed. We illustrate the effect this has on the precision of reported estimates in figure 2, which illustrates that with as few as 30 expected tasks, reporting a precision of ±20.5% is possible.

As this work is a first of its kind (globally as far as we are aware), we feel that providing estimates of the proportion of tasks done/left undone with a precision of approximately ±20% will be sufficient to provide valuable insights into the challenges faced by nurses in providing newborn care.

## Procedures

For each newborn being observed, the diagnosis and disease severity information will be collected from medical records as this informs the expected number of tasks. At the beginning of every shift for which observations will be made, the total number of nursing tasks expected to be delivered will be determined with reference to the medical and nursing records (for disease severity and specific interventions like phototherapy) and general aspects of care each baby should receive. For each of the newborns selected to participate in the study, we will make direct observations on how often certain routine nursing tasks (listed in table 1) are undertaken in a 12-hour shift (07:00–19:00 hours or 19:00–07:00 hours) using an observation checklist. Observations will be stopped if a baby is transferred out of a section, changes condition and becomes critically ill (requires specialised treatment which the minimum draft nursing standards for neonatal

| Table 1 | Routine and critical tasks for observation |
|---|---|
| **Routine tasks** | **Critical tasks** |
| Patient assessment at the beginning of each shift<br>Cleaning of the baby<br>Changing baby's linen<br>Changing the baby's position<br>Checking incubator settings<br>Ward round attendance and active note taking<br>Weighing<br>Elimination care<br>Communication/counselling parents<br>Cord care<br>Vital signs measurement<br>► Pulse rate<br>► Temperature<br>► Respiratory rate<br>► Oxygen saturation<br>Documentation<br>► Updating the nursing cardex<br>► Discharge and admission registration | Nasogastric feeding<br>► Insertion of the nasal gastric tube (NGT)<br>► Testing whether it is in the correct position<br>► Checking for gastric aspirate before feeding<br>► Preparation of feeds and counterchecking feed volumes<br>► Actual feeding and charting the feeds<br>► Intravenous drug/fluid administration<br>► Reviewing treatment sheet<br>► Checking cannula sites—care<br>► Regulating flow<br>► Input/output charting for fluids<br>► Document treatments given<br>Oxygen therapy<br>► Fixing of oxygen/nasal prongs<br>► Checking tube position and nostril—care, damage<br>► Initiating and regulating oxygen flow<br>► Documenting oxygen treatment<br>Photo therapy<br>► Baby positioning<br>► Placing/checking eye pad is in place<br>► Checking eyes for damage<br>► Checking and monitoring phototherapy settings<br>► Documenting of phototherapy<br>Support for kangaroo mother care (KMC) |

care do not apply) or discharged, however, we will use the data collected up to the point of exit and the number of observation hours will be documented. If the babies' condition changes and their category changes but they remain in the same observation area, we will document this change and revise the expected number of tasks. In both instances, the effective denominator for expected nursing tasks will be changed as required.

The observer will be stationed in a ward area where they do not obstruct care provision but can observe the care being provided to the newborns selected for observation. Because most of the documentation activities happen at the nurses' desk/station, all tasks related to documentation cannot be observed from the cot side. The documented tasks for the babies under observation will be evaluated at the end of each 12-hour shift for which observations will be made. We acknowledge that 12 hours is a long period and the observer's efficiency for making observations might reduce as time progresses within an observation time block and study period. We will factor in rest periods in the 12-hour shifts that coincide with when nurses take their breaks, for instance, during tea and lunch breaks, periods which we anticipate limited or no tasks will be undertaken. Further, to allow enough rest between observation time blocks, we will aim to have a maximum of three, 12-hour observation periods per week per observer with at least a 24-hour rest period between observations. The 12-hour periods were selected because they span nursing shift change overs that allowed documentation of care round-the-clock and made random selection of time blocks more feasible.

The observation checklist (online supplementary file 1) is based on nursing standards produced after two expert group meetings held in November 2015 and July 2016 at the Kenya Medical Research Institute Wellcome Trust Research Programme offices. The nurse stakeholder group comprised expert nurses who are in the teaching profession or senior practitioners and including the acting chief nursing officer for Kenya. They defined a minimum standard for performing nursing tasks on newborn units.[28] A subset of tasks identified as critical (listed in table 1) by the nurse stakeholder group was explicitly marked as to only be performed by nurses and not by other personnel due to the skills required when they are delivered. Where necessary, tasks are broken down into manageable observable task components to facilitate observation. For instance, nasogastric feeding is broken down into insertion of the NGT (as required), checking for gastric aspirate before feeding, counter checking feed volumes to be given with the feeds prescription, actual feeding and charting the feeds given. The purpose of observation is not to assess how well any particular aspect of a task is done (eg, the care taken in administering nasogastric feeds) but simply to determine if the task (or task component) was (or was not) done at all.

Prior to the start of the study, the observation checklist (online supplementary file 1) will be extensively piloted over a period of 6 weeks to determine the quantity and quality of data that can be reasonably gathered and will be adapted as needed. The tool will be piloted in one public health facility that will not be used as a study site in the final study by the research assistant who will

subsequently be responsible for training the data clerks. During piloting, we will observe care provision in each of the nursing 12-hour time blocks to explore what tasks can be observed, what number of newborns will be logistically feasible to observe, the different nursing routines in the different shifts and the documents used for reporting on nursing activities.

For purposes of this study, we adopted the definition of missed care reported in wider literature,[22 31] and was therefore defined as care that the nursing advisory group regard as necessary (primarily essential neonatal nursing tasks) as part of routine newborn care that are left undone or are delivered by any person other than the nurse or a qualified healthcare provider (nutritionist, doctor, clinical officer, etc). However, tasks that will be done by a senior student nurse under direct observation/supervision by the nurse will be considered as done and documented as not done if no supervision is observed. For example, tasks done by a senior student nurse who is being supervised to conduct nasogastric feeding by a qualified nurse in the room at the time and focused on the supervision will be regarded as done. If the same student does the task while the nurse supervising is in a different part of the ward, it will be regarded as not done (with a record of who completed the task made). Additional data to be collected related to each block of observation time will include data on the nurse-to-patient ratio for the ward as a whole, what additional staff are present within the unit, for example, nursing students, support staff, etc and patient workloads within the different sections of the ward.

### Practical considerations for undertaking direct observations

Observations will be made by a person familiar with the hospital environment, equipment, processes (like ward rounds) and language but who is not a nurse or clinician (doctor/clinical officer). Given the potential sensitivity of this form of observation, it is important that this person is considered a professional rather than an 'outsider' (who might not be bound by professional codes of confidentiality). Having an observer who has an understanding of the setting but who is not a clinician or nurse may also overcome problems of the observer making judgements about what is being observed based on their own standards of practice or being influenced as they make observations by professional allegiances (eg, a nurse observer may not wish to record that a task is left undone). There might also be ethical challenges for an observer with a clinical/nurse background who might feel obliged to intervene in the provision of care or become co-opted to complete tasks. An observer who is not licensed to offer the form of interventions/care being given to babies will still, however, be able to report to healthcare providers within the unit in the event they identify gaps or situations in which newborns are put at risk. For instance, if they see a newborn convulsing or vomiting, they will alert the nurse. Suitable backgrounds for observers might include nutritionists, health records officers, ward-based

clerical workers, or laboratory or pharmacy staff with at least 2 years experience within hospital settings. In this study nutritionists will be recruited to undertake the observations.

We will recruit six observers who meet the above criterion. The observers will be trained for a period of 2 weeks by one of the study personnel who has a nursing background and has 2 years experience providing care to newborns within a newborn unit. Standard operating procedures (SOPs) will be developed and will serve as a guide for the observational work. During the training period, the observers and the trainer will do observations and comparisons will be made. Any differences will be discussed and where necessary the SOPs will be revised to improve on clarity. Supervision during the data collection period will be undertaken by study staff with a nursing background with routine weekly reviews of the observation checklists for completion and consistency.

To reduce the Hawthorne effect where nurses might change the way they provide care when they are being observed, the observer will spend at least 1 week in the health facility before starting the observational work. This will also allow him/her to familiarise themselves with the environment, to explain the study to staff and parents and for the staff within the newborn unit to get used to them. In addition, the observer will make it clear that the observations are not an assessment but a means of understanding what care is possible to provide given existing resource constraints. Formal study observations will then begin in the period after the 1-week familiarisation period. This approach is supported by evidence that healthcare providers change their practice slightly when the observations start but these changes are short lived and quickly dissipate with healthcare providers soon reverting to their previous practice.[32 33] In addition, the 12-hour observation periods are randomly generated and the hospitals will not be aware of the day, shift or category of babies the observer would next come to do observations for.

### Practical and public involvement

Patients and the public were not involved in the development of the research question or outcome measures. However, as part of the broader set of work within which this study is embedded, there is ongoing work to understand patient experiences focusing on experiences of mothers with newborns admitted within the newborn unit. We hope findings from our study will complement those from the patient experiences work and will provide insights on aspects of care that are important to mothers and inform the design of interventions to improve care.

### Data management and analysis

Data will be collected on paper-based observation checklists, one for each baby that is the focus of observation. No participant identifiable data will be collected and data collection instruments will be identified only by a unique study identifierallocated to each baby. The place, date

and time of the shift will also be anonymised by using only specific codes for observation shifts at data entry. Data on the paper-based observation checklists will be checked for completeness by a supervisor at the end of each day. The checklists will be double entered into a custom-made database using Research Electronic Data Capture (REDcap) with in-built range and consistency checks. The entered data will be checked at the end of each day using precoded scripts for entry errors and completeness. Data will be exported for cleaning and analyses in Stata V.13 (Stata).

Descriptive analysis will be undertaken on the pooled data across hospitals to determine the overall prevalence of care left undone and the common tasks that are left undone. We will also report the average number of tasks left undone per newborn and the common tasks left undone per newborn. Secondary analysis will be undertaken to explore variations in care done (or left undone) by the various sub-categories that will include: sector, nursing shift, nurse-to-patient ratios and category of the baby (disease severity). Missing or incomplete data will be coded as a category and where necessary presented as such. When reporting on effective number of tasks done (or left undone), missing/incomplete data will be excluded from the effective denominator the task would have contributed to and hence avoid spurious inflation of the denominator.

## Ethical considerations

The focus of our direct observations of care is what happens to newborns and we will not record any names or other identifying features of people who may be providing care to these newborns. We will only report pooled results stratified by sector, nurse-to-patient staffing ratios and category/severity of disease so as to preserve the confidentiality of the health facilities.

During the 1-week familiarisation period, we will seek written individual informed consent from all nurses who will be providing care within the newborn unit. Additional consent forms and study briefs will be left in the ward to allow nurses not available during this introductory period but who provide care in the newborn unit to review and indicate their willingness to participate. During an observation shift, these nurses will be asked for individual informed consent before the start of the observations. Additionally, at the beginning of every shift for which direct observations will be made, group verbal informed consent will be sought from nurses after an explanation of the study has been made (this is in addition to the 1-week familiarisation during which the study will be explained). We will also provide printed study briefs targeting healthcare providers explaining the study and indicating that they are free to decline from being observed as an approach to ongoing consent.

Written informed consent will be sought from the mothers of babies considered for direct observations at the start of each observation period. In Kenya, pregnant adolescents between ages (15 and 17 years) are considered 'emancipated minors' and their written informed consent will be obtained.[34–36] The start of the direct observation shift (07:00 or 19:00 hours) is just afterwhen mothers are in the newborn unit for the 3 hourly feeding session at 06:00 or 18:00 hours, as such mothers will be approached during this period. In instances, where mothers indicate more time is required to make a decision or to consult they will be allowed to do so and another mother whose baby meets the criteria for observation will be approached. In cases where nurses or parents decline consent, babies under their care will not be considered for direct observation. It will be made clear that at any stage nurses or parents can withdraw consent/permission for observation, temporarily or for the rest of a shift, without explanation and with no penalty. Observations will only be undertaken where both the mother and nurses have provided consent to the study, respectively.

Ethical permission to undertake the study in the respective hospitals was sought from each of the hospital's administrative offices.

## Dissemination of findings

An expert advisory group, including partners from the Ministry of Health, Nursing Council of Kenya, Nairobi City County, Kenya Medical Training College, Kenyatta University and hospitals providing inpatient neonatal services has been involved in the development of the standards of nursing care being used to understand missed care in this study. This expert group has provided support for the study and is in itself a key consumer of the results as these members are directly involved in policy-making and training. This group will also provide a key channel for wider dissemination of the findings.

Highlighting the extent and magnitude of care left undone will provide important insights on the nursing care available to newborns admitted within newborn units, highlight human resource issues warranting attention and will likely influence recommendations on staffing norms and how care is organised and delivered in newborn units. Further, findings from this work will help guide the design of approaches and interventions for improving facility-based care for this highly vulnerable population.

At the end of the study, findings will be provided to and discussed with participating hospitals and other relevant stakeholders. More widely, the international scientific community will be targeted via publications in peer-reviewed journals as well as conferences.

## Global public health relevance

Facility-based care has been cited as one key approach to reduce neonatal mortality if evidence-based, high-impact, low-cost interventions are appropriately delivered at high coverage.[4] Globally, improving quality of newborn care provided by facilities might save 600 000 small and ill neonates annually[4] while data from Kenya suggest that by 2030, 6000 newborn lives could be saved through the

provision of childbirth and newborn care intervention packages alone.[37]

The Kenyan government is promoting facility-based delivery by making maternity care free for women. This has resulted in an increase in utilisation of maternal and newborn health services, potentially increasing the number of newborns accessing care in health facilities. At the same time, significant challenges exist in nurse staffing and availability of wider resources[27] and this may limit any impact of increased access to care. The extent of missed nursing care in newborn units in LMIC and how it impacts on quality of care delivered has not previously been described. This is despite care in the newborn unit being heavily dependent on nurses. By characterising care left undone, we will identify important potential gaps in care delivered within newborn units that can inform discussions on how best to address these gaps and improve quality of care in Kenyan hospitals.

This study will be the first attempt, of which we are aware, to develop and apply direct observational missed care tools to understanding neonatal nursing care provision in LMIC. In fact, a recent review by Jones et al identified that only questionnaires have previously been used to quantify missed care and these might be limited by reporting bias.[22] However, we acknowledge that direct observational methods have limitations on the number and actual tasks that can be observed, might be influenced by observer bias and are at risk of Hawthorne effect. As such, the tasks in our observation checklist (online supplementary file 1) are limited to essential neonatal nursing tasks provided at the bedside that can be observed while those linked to documentation will involve review of medical records at the nurses' desk since documentation of care is done at the nurses' desk and not at the cot side. Moreover, observing the documentation of a task or reviewing records for evidence of documentation is likely to provide similar results. An alternative method for data collection that we considered was the use of video in the newborn unit with later evaluation of what is done (or not done). However, informal discussions indicated this might be controversial at this stage due to the ethical and medicolegal issues that might emanate from this approach and the administrative approvals required for video recording in hospitals, nonetheless this is a potential area for future research. The selection of health facilities was purposive, as support by the hospital administration was important due to the nature of data collection and the sensitivity associated with direct observation of care. As such, we cannot rule out selection bias as facilities providing better care might have been more likely to agree to partake in the study. However, this is the first such study using direct observational methods to quantify missed care and in a LMIC and therefore the findings will still be important. We anticipate that this study will, therefore, be of global interest and provide an opportunity for application of the methods developed to other neonatal settings or in other disciplines to better understand nursing care provision and quality gaps. Further, findings of this work will be important in contributing to thinking on nurse staffing norms and how care is organised and delivered in LMIC newborn units. This will be crucial in influencing longer term policy on human resource planning. Most directly the findings will feed into the 'Kenya Task Sharing Policy and Guidelines for Health Care Services',[38] an initiative being led by Ministry of Health alongside other stakeholders as one of the ways of tackling health workforce shortages.

**Contributors** DG, ME and GAVM designed the study with contributions from RK and ET. DG, NA and GS were responsible for the coordination and supervision of data collection. DG wrote the study protocol with substantial and critical input from all coauthors. All authors read and approved the final version of the manuscript.

**Funding** This work was supported by the Health Systems Research Initiative joint grant provided by the Department for International Development, UK (DFID), Economic and Social Research Council (ESRC), Medical Research Council (MRC) and Wellcome Trust, grant number MR/M015386/1. ME is supported by a Wellcome Trust Senior Research Fellowship (#097170).

**Competing interests** None declared.

**Patient consent** Not required.

**Ethics approval** Ethical approval to conduct this study in all hospitals has been granted by the Kenya Medical Research Institute (KEMRI) Scientific and Ethics Review Unit (Approval No. KEMRI/SERU/CGMR-C/065/3404).

**Provenance and peer review** Not commissioned; externally peer reviewed.

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
