## [Reviewer comments · BMJ Open]

ARTICLE DETAILS

TITLE (PROVISIONAL)	Quantifying nursing care delivered in Kenyan newborn units: Protocol for a cross-sectional direct-observational study
AUTHORS	Gathara, David; Serem, George; Murphy, Georgina; Abuya, Nancy; Kuria, Rose; Tallam, Edna; English, Mike

VERSION 1 – REVIEW

REVIEWER	Marjan Ahmad Shirvani Mazandaran University of Medical Sciences, Iran
REVIEW RETURNED	23-Feb-2018

GENERAL COMMENTS	In this study, missed care in newborn units which plays an important role in neonatal mortality, will be considered. Some more information are needed to add for more clarification. Title It is necessary to add the place and date of study in the end of the title. Abstract Methods: kindly report the date of the study, sample size, method of sampling, and also describe study population, method and instrument of data gathering and statistical tests. Ethics: to receive informed consent and anonymous report about caregivers must be considered. Strengths and limitations “However, our scope is limited by the few number of newborn to be observed within each hospital”. This limitation could be prevent by revising the estimation of the sample size of newborns I think the most limitation of the study is the Hawthorne effect. Although you explained about it, you cannot completely omit it. Methods and analysis Please report the place and date of study in the first paragraph. Study site (geographical) 24/7 is not clear in this phrase: “24/7 inpatient newborn care” Step 2 Kindly mention the method of random sampling for twelve shifts/time blocks Please write “more than one section” instead of “>1 section” Procedures You said because most of the activities happen at the nurses’ desk/station, all tasks cannot be observed. It is a considerable defect and must be managed by observing the nurses’ desk or other approaches. References All references have not reported with the same pattern. Furthermore, Information about some references need to be completed. Kindly, follow the instruction of BMJ Open for writing references.
---

REVIEWER	Hazel A Smith Paediatric Intensive Care Unit, Our Lady's Children's Hospital, Crumlin, Dublin 12, Ireland
REVIEW RETURNED	27-Feb-2018

GENERAL COMMENTS	Title of paper: Quantifying nursing care delivered in newborn units: Protocol for a direct-observational study Manuscript ID: bmjopen-2018-022020 General points  1) Is it missed care due to staffing, training, resources, time....?As depending on why the care is being missed will affect the recommendations of the study 2) When taking about nursing care, nursing standards etc I would also have 'neonatal' in front. For example, in Strength and Limitations: 'new approaches to quantifying neonatal nursing care...' As the care given by a neonatal nurse, to a neonate, can be different to a paediatric and general nurse. 3) I would ask someone not involved in the study to read the manuscript for grammar checks. For example, in Strengths and Limitations (page 3/17, lines 22-24) "The different sectors (public, private-for-profit, private-not-for-profit) in health service provision have included in this study." 4) Consent: How will under-age parents be consented? Only some nurses will be providing their consent prior to the study starting and parental consent will also be sought at the point when the observer arrives at the hospital. The general rule of thumb is that parents/ staff have time to read/ discuss the study information and given space to think about it. How is the study addressing the fact that parents & some staff will not have time to discuss the study with other people and may feel pressured to saying 'yes' as the observer has already arrived. 5) Your observer will be working a more than 12hrs (as they will need to arrive prior to the nursing shift starting and need to remain on site to review medical charts once the nursing shift is completed). This is a big ask, and will also mean that during their breaks etc they will not be in a position to observe nursing tasks. How will the study manage this? 6) Be consistent. (1) Is it 'missed care' or 'care done' that you are exploring? I would use the same language for your
---

outcomes throughout. (2) Are you including 'sick or preterm newborns' or just 'sick newborns'? (3) For some facts you write the statistics starting at the highest number and at other times you start with the lowest, for example "1.2 to 0.008 per 1000 population" and "0.21 to 0.40 per 1000 population". (4) The aim of the study is 'is to quantify the nursing care' but in the paper you write about how the study will give information on the 'quality of care'. Quality describes how well a function is done and quantify is to express the quantity. Are you measuring both?, and if so, how are you measuring both?

Abstract

- 1) Page 2/17, lines: 47-48: Methods and analysis – 'the aim of this study is to quantify the nursing care'. If the word count allows I would be more specific, is it quantify all possible care i.e. physical and psychology or essential nursing neonatal care or....? By stating what care you are referring to it is clear to the reader what tasks (missed care) you are referring to.
- 2) In the abstract you said that you will describe any sharing of tasks with people not formally qualified to provide care. How do you know they are not formally qualified; is it non-hospital staff (as family members etc could be doctors or nurses) or will you include hospital staff who are undertaking tasks they are formal not qualified to do so?

Ethics

- 1) Does the ethical approval provided for the study cover all hospital sites that you are visiting?

Need to state here if it does and if it doesn't that you are applying for ethical approval where needed.

Strengths and Limitations

- 1) (This links in with my first point under general comments). You appear to already know the answer to your question. In the last point you state that this study will be highlighting human resource issues that warrant attention and will

influence recommendations on staffing norms. Missed care could be the result of human resources issues but the concern could be with the availability of staff, with the required training, and not staffing recommendations. And if it is due to the availability of staff then your recommendations will need to take account of the nursing staff that are available to work in newborn units.

Introduction

- 1) Page 3/17, lines 40-41: "As a consequence about 45% of child mortality is neonatal mortality[1]" I would state that the 45% of child mortality is for children under 5 years of age.

Methods and analysis

- 1) Page 4/17, lines 55: Need to clarify if the two health facilities that have declined to participate in other studies are being excluded and how this exclusion could impact on your findings.
- 2) Reference where you are getting your figures from.
 - 34 facilities (100% of 24/7 care)
 - 3 excluded (*so these three units combined provide 1% of 24/7 care?*)
 - 31 facilities (99% of 24/7 care)
 - 13 facilities with an admission rate of >100 newborns (96% of 24/7 care)
 - So that means that 18 facilities, with 24/7 care, account for 3% (99%-96%) of admissions? And for the 13 selected facilities that provide ~7.4% of 24/7 care each?
- 3) Need more information on how and why the six centres were 'purposefully selected'. Need to ensure that selection bias is not taking place.
- 4) Page 6/17, lines 25-32: Step 3. Without doing a pilot how to you know this is feasible and what is your back up plan?

5) For sample sizes and estimates I have requested that a statistician reviews this section, as I have concerns about the sampling but would value the opinion of a statistician. Also, you have not mentioned that you have received any statistically support/ advice for this protocol and I would advise seeking their guidance.

6) Page 7/17, Procedures:

- You are extracting medical information at the start of the shift – when is parental consent being sought?
- Determining the total number of tasks at the start of the shift could be time consuming for non-nursing observers. What happens if the observer misses tasks as they are going through the notes?
- Lines 27: you state the 12 hour shift under observation is from 7am to 7pm but if you are including nights than shouldn't this be 7am-7pm or 7pm-7am?
- Routine and critical tasks can differ based on day or night shift how are you accounting for this?
- Revising expected tasks if a child changes area also takes time and this will mean that the observer will not be able to collect data. It could also mean depending on where the child is transfer to within the unit that the observer will not be able to collect data any more on that child.
- How will you handle incomplete data – if the observer can no longer collect data on a child due to being discharged, transferred out of unit or to another area within the hospital? How does this impact on estimates and sample sizes etc?
- Page 8/17, lines 13-16. You discuss piloting the study – I would include a lot more information on this, as how the pilot is done will affect how the study is run.
- Table 1: how frequently are you expecting each of the routine and critical tasks to be done, at a minimum?

References

1) Not all references are typed out correctly, for example:

Page 13/17, lines: 5: "1. WHO U. A Decade of Tracking Progress for Maternal, Newborn and Child Survival. 2015."

REVIEWER	John Harris University of Pittsburgh, USA
REVIEW RETURNED	27-Feb-2018

GENERAL COMMENTS	The authors present a study protocol for a cross-sectional study of nurses using structured observations of newborn nursing care to quantify the amount of "missed care" (appropriate nursing care not provided) in a lower-middle income country. The protocol does a good job of establishing the need to improve newborn care in low and middle income country healthcare. Overall, the description of the study is thorough and detailed to adequately inform a reader. The description of sampling methodology and sample size calculations are especially clear. Comments:  1. Further description of why the study method was chosen would be helpful. Structured observations is a reasonable methodology, but it would be helpful to contrast it with other methods including surveys, simulated patient exercises, video observation, semi-structure interviews, focus groups, etc. 2. "Missed care" has at least two etiologies: process inconsistency (care can't be provided because of interrupting tasks, inadequate staffing, etc) and inadequate knowledge or training (so care is not provided because the provider doesn't realize how to do it properly). The present methodology will group both problems into one episode of "missed care". The methodology doesn't have an obvious way to differentiate between these two problems. 3. Beyond collecting information about the baby's condition and the checklist, is there any other information being collected? Qualitative observations noting the chaos, organization, methods the nurse used to complete tasks, experience of the nurse, training/background of the nurse, how the nurse interacted with other staff, etc would be very helpful. 4. An exact checklist was not provided, but hopefully it goes into enough detail to give some sense of whether a complex task like "Communication/counselling parents" is done in an appropriate or inappropriate manner. More information about how the checklist was created would be of interest to the reader.
---

VERSION 1 – AUTHOR RESPONSE

Reviewers' Comments to Author:

Reviewer: 1

Reviewer Name: Marjan Ahmad Shirvani

Institution and Country: Mazandaran University of Medical Sciences, Iran

Competing Interests: None declared

In this study, missed care in newborn units which plays an important role in neonatal mortality, will be considered. Some more information are needed to add for more clarification.

Title

It is necessary to add the place and date of study in the end of the title.

Thank you for this comment. We have now revised the title to include the setting and study-design. The study dates have been included in the methods section

“Quantifying nursing care delivered in Kenyan newborn units: Protocol for a cross-sectional direct-observational study”

Abstract

Methods: kindly report the date of the study, sample size, method of sampling, and also describe study population, method and instrument of data gathering and statistical tests.

The abstract methods section has now been revised to include a summary of the above details due to the word limit allocated to this section.

“The aim of this study is to quantify essential neonatal nursing care provided to newborns within newborn units. We will undertake a cross-sectional study utilising direct-observational methods within newborn units in 6 health facilities in Nairobi City County across the public, private-for-profit and private-not-for-profit sectors. A total of 216 newborns will be observed between 1st September 2017 and 30th May 2018. Stratified random sampling will be used to select random 12-hour observation periods while purposive sampling will be used to identify newborns for direct-observation. We will report the overall prevalence of care left undone, the common tasks that are left undone and describe any sharing of tasks with people not formally qualified to provide care.”

Ethics: to receive informed consent and anonymous report about caregivers must be considered.

We have now included a statement on informed consent. Detailed ethics and consent procedures are provided in the main manuscript due to the word limit allocated to this section

“Written informed consent will be sought from mothers and nurses.”

Strengths and limitations

“However, our scope is limited by the few number of newborn to be observed within each hospital”. This limitation could be prevent by revising the estimation of the sample size of newborns

I think the most limitation of the study is the Hawthorne effect. Although you explained about it, you cannot completely omit it.

We thank the reviewer for this comment and acknowledge that indeed we cannot fully eliminate the effect of Hawthorne effect despite our efforts to minimize it. We have now included this as part of our limitations

“Despite our efforts to minimize the Hawthorne effect, we cannot rule out the possibility of nurses changing the way they provided care during the observational periods”

Methods and analysis

Please report the place and date of study in the first paragraph.

Study site (geographical)

We have now revised this section to include the study site and dates for the study.

“This is a cross-sectional study that will involve direct observation of care provided to individual newborns with the aim of describing the essential neonatal nursing care given and missed within newborn units. It will be undertaken in 6 hospitals in Nairobi City County, Kenya, in the period 1st September 2017 to 30th May 2018”

24/7 is not clear in this phrase: “24/7 inpatient newborn care”

This statement has now been revised and explained to improve on clarity

“This collaboration includes work to characterise all the facilities providing inpatient newborn care 24 hours, 7 days a week (hereafter referred to as 24/7) in Nairobi and ethnographic work to understand the wider context and practice of neonatal nursing.”

Step 2

Kindly mention the method of random sampling for twelve shifts/time blocks

Please write “more than one section” instead of “>1 section”

Thank you for this comment. We have now updated the sampling method.

"In each hospital, stratified-random sampling will be used to generate a random sample of twelve shifts/time blocks of 12 hours (144 observation hours per hospital) from within a 3-week period stratified by disease severity (category A, B, C), weekdays and weekends as well as night and day shifts as care has been shown to vary across these periods"

Procedures

You said because most of the activities happen at the nurses' desk/station, all tasks cannot be observed. It is a considerable defect and must be managed by observing the nurses' desk or other approaches.

Thank you for this comment. The text in the manuscript refers to documentation activities. We acknowledge this is a limitation and we have now highlighted this in the manuscript. The focus of our work is on delivery of 'bedside nursing care' and we acknowledge that documentation is an important part of care but is not the focus of study. However, observing the documentation of a task or reviewing records for evidence of documentation is likely to provide similar results. In fact, nurses often record a cluster/group of activities and do not document each task after completion.

"However, we acknowledge that direct observational methods have limitations on the number and actual tasks that can be observed, might be influenced by observer bias and are at risk of Hawthorne effect. As such, the tasks in our observation checklist are limited to essential neonatal nursing tasks provided at the bedside that can be observed while those linked to documentation will involve review of medical records at the nurses' desk since documentation of care is done at the nurses' desk and not at the cot-side. Moreover, observing the documentation of a task or reviewing records for evidence of documentation is likely to provide similar results."

References

All references have not reported with the same pattern. Furthermore, information about some references need to be completed. Kindly, follow the instruction of BMJ Open for writing references.

The references have now been corrected and updated.

Reviewer: 2

Reviewer Name: Hazel A Smith

Institution and Country: Paediatric Intensive Care Unit, Our Lady's Children's Hospital, Crumlin, Dublin 12, Ireland

Competing Interests: None declared

Many thanks for giving me this opportunity to review your paper. The reason that I have said to reject is that you have not yet undertaken your pilot study.

On page 7 you state: 'Prior to the start of the study, the observation checklist will be extensively piloted over a period of 6 weeks to determine the quantity and quality of data that can be reasonably gathered and will be adapted as needed'. As you said this study is a first of its kind, so use the pilot experience and findings to develop your study protocol. The study question is valid and will provide valuable information but at the moment I feel there are too many vague descriptions in the protocol to recommend for publication. Which I feel is a reflection that the protocol has not been developed based on the results from the pilot study.

We would like to clarify that a pilot study was undertaken further refinement of the data collection methods and study tools. We have provided additional information on activities undertaken during the pilot phase and how they influenced the study.

"Prior to the start of the study, the observation checklist will be extensively piloted over a period of 6 weeks to determine the quantity and quality of data that can be reasonably gathered and will be adapted as needed. The tool will be piloted in one public health facility that will not be used as a study site in the final study by the research assistant who will subsequently be responsible for training the data clerks. During piloting, we will observe care provision in each of the nursing 12-hour time blocks to explore what tasks can be observed, what number of newborns will be logistically feasible to observe, the different nursing routines in the different shifts and the documents used for reporting on nursing activities"

Another point to note is that the BMJ state: 'Protocol papers should report planned or ongoing studies. The dates of the study should be included in the manuscript'. You have not included any dates in your paper.

These details have now been included in the methods section. While the data collection might be almost complete, the paper was first submitted in January 2018.

"This is a cross-sectional study that will involve direct observation of care provided to individual newborns with the aim of describing the essential neonatal nursing care given and missed within newborn units. It will be undertaken in 6 hospitals in Nairobi City County, Kenya, in the period 1st September 2017 to 30th May 2018"

I have written some additional points in a word document, please see attached.

General points

1) Is it missed care due to staffing, training, resources, time...?As depending on why the care is being missed will affect the recommendations of the study

The aim of our study was to quantify routine neonatal nursing care tasks delivered to newborns admitted within newborn units. As such, the delivery of these tasks is unlikely to be influenced by training. However, we acknowledge that the care missed might be influenced by staffing levels and availability of time. We are collecting data on the number of nurses on duty and sick newborns in the unit during the observation shift and will therefore quantify if these are some of the contributors to missed care. Additionally, this study is part of a wider body of work that examines staffing in more detail and seeks to provide important data to help understand the consequences of poor staffing (in press and now referenced in the main manuscript).

"This collaboration includes work to characterise all the facilities providing inpatient newborn care 24 hours, 7 days a week (hereafter referred to as 24/7) in Nairobi, quality of clinical care provided newborns, and ethnographic work to understand the wider context and practice of neonatal nursing."

2) When taking about nursing care, nursing standards etc I would also have 'neonatal' in front. For example, in Strength and Limitations: 'new approaches to quantifying neonatal nursing care...' As the care given by a neonatal nurse, to a neonate, can be different to a paediatric and general nurse.

Thank you for this suggestion we have revised the manuscript to take account of this recommendation.

3) I would ask someone not involved in the study to read the manuscript for grammar checks. For example, in Strengths and Limitations (page 3/17, lines 22-24) "The different sectors (public, private-for-profit, private-not-for-profit) in health service provision have included in this study."

Thank you for this suggestion. We have now revised the manuscript and checked the manuscript for typographical and grammatical errors.

4) Consent: How will under-age parents be consented? Only some nurses will be providing their consent prior to the study starting and parental consent will also be sought at the point when the observer arrives at the hospital. The general rule of thumb is that parents/ staff have time to read/ discuss the study information and given space to think about it. How is the study addressing the fact that parents & some staff will not have time to discuss the study with other people and may feel pressured to saying 'yes' as the observer has already arrived.

Thank you for this comment. For the nurses, the data collection assistants will spend one week in the hospital prior to the formal start of the study with the hope that during this period they will meet and consent over 90% of the nurse providing care in the different nursing shifts. We also provide information sheets and additional consent forms within the ward for nurses to read when they have free time especially those who might have not been present when the study was explained. We have revise the text in the manuscript to improve on clarity

“During the one-week familiarisation period, we will seek written individual informed consent from all nurses who will be providing care within the newborn unit. Additional consent forms and study briefs will be left in the ward to allow nurses not available during this introductory period but who provide care in the newborn unit to review and indicate their willingness to participate. During an observation shift these nurses will be asked for individual informed consent before the start of the observations. Additionally, at the beginning of every shift for which direct observations will be made, group verbal informed consent will be sought from nurses after an explanation of the study has been made (this is in addition to the 1-week familiarisation during which the study will be explained).”

For the mothers, the observational periods are preceded by the 3-hourly feeding sessions (6am or 6pm) before the 7am time-point when we start the observations. This provides adequate time to get consent. If a mother indicates they need more time to decide or consult, this is allowed for and another mother is considered for inclusion into the study. In Kenya, pregnant adolescents between ages (15-17 years are considered ‘emancipated minors’ and their written informed consent was obtained. We have revised the text to improve on clarity.

“Written informed consent will be sought from the mothers of babies considered for direct observations at the start of each observation period. In Kenya, pregnant adolescents between ages (15-17 years are considered ‘emancipated minors’ and their written informed consent was also obtained. The start of the direct observation shift (7am or 7pm) is just before when mothers are in the newborn unit for the 3-hourly feeding session at 6am or 6pm, as such mothers will be approached during this period. In instances, where mothers indicate more time is required to make a decision or to consult they will be allowed to do so and another mother whose baby met the criteria for observation will be approached. Further, the observer will be located in the same areas for 12 hours and the mothers will be encouraged to seek as much information on the study over this period and it will be emphasized that they can withdraw from the study at any point without penalty”

5) Your observer will be working a more than 12hrs (as they will need to arrive prior to the nursing shift starting and need to remain on site to review medical charts once the nursing shift is completed). This is a big ask, and will also mean that during their breaks etc they will not be in a position to observe nursing tasks. How will the study manage this?

Thank you for this comment. We acknowledge that the observer would be in the ward for long hours. The 12-hour periods were selected because they span nursing shift change overs that allowed documentation of care round-the-clock and made random selection of time blocks more feasible. Additionally, to reduce Hawthorne effect where hospitals plan to provide better care when they anticipate we will be doing observations e.g. by increasing number of nursing staff in a shift we limited consenting of mothers to the day of the observations. We are allowing the observer a full day’s rest before they get onto the next observational shift with a maximum of 3 shifts in 7 days but also encouraged. We have now clarified this in the text.

“We will factor in rest periods in the 12-hour shifts that coincide with when nurses take their breaks, for instance, during tea and lunch breaks, periods which we anticipated limited or no tasks will be undertaken. Further, to allow enough rest between observation time blocks, we will aim to have a maximum of 3, 12-hour observation periods per week per observer with at least a 24-hour rest period between observations. The 12-hour periods were selected because they span nursing shift change overs that allowed documentation of care round-the-clock and made random selection of time blocks more feasible.”

6) Be consistent.

(1) Is it ‘missed care’ or ‘care done’ that you are exploring? I would use the same language for your outcomes throughout.

We are interested in how much essential neonatal nursing care gets done and will therefore identify what is missed by comparing what is done with what is supposed to be done. We have revised the language in the manuscript to improve on clarity

(2) Are you including ‘sick or preterm newborns’ or just ‘sick newborns’?

We are interested in all newborns admitted in the newborn unit both ‘sick newborns’ and pre-term babies who might be in the unit to gain weight or sick newborns who have recovered and

are completing their antibiotic medication. We revised text in the manuscript to reflect this population of interest.

(3) For some facts you write the statistics starting at the highest number and at other times you start with the lowest, for example “1.2 to 0.008 per 1000 population” and “0.21 to 0.40 per 1000 population”.

We have now revised the text to ensure consistency

“In Kenya, Wakaba and colleagues reported that public sector nursing densities ranged between 0.008 to 1.2 per 1000 population across counties compared to an internationally suggested minimum health workforce threshold of 2.5/1000 population for doctors, nurses and midwives. In Nairobi County, the nurse densities ranged between 0.21 to 0.40 per 1000 population”

(4) The aim of the study is ‘is to quantify the nursing care’ but in the paper you write about how the study will give information on the ‘quality of care’. Quality describes how well a function is done and quantify is to express the quantity. Are you measuring both?, and if so, how are you measuring both?

We have rephrased the statement to indicate findings will highlight the quantity of nursing care delivered.

“Highlighting the extent and magnitude of care left undone will provide important insights on the nursing care available to newborns admitted within newborn units, highlight human resource issues warranting attention and will likely influence recommendations on staffing norms and how care is organised and delivered in newborn units.”

Abstract

1) Page 2/17, lines: 47-48: Methods and analysis – ‘the aim of this study is to quantify the nursing care’. If the word count allows I would be more specific, is it quantify all possible care i.e. physical and psychology or essential nursing neonatal care or....? By stating what care you are referring to it is clear to the reader what tasks (missed care) you are referring to.

Thank you for this comment. We have now revised the manuscript to reflect essential neonatal nursing care with a focus on largely the technical aspects of care that are most feasibly observed. We have also included the observation checklist as supplementary material to illustrate the neonatal nursing tasks we are observing.

“This is a cross-sectional study that will involve direct observation of care provided to individual newborns with the aim of describing the essential neonatal nursing care given or missed within newborn units.”

2) In the abstract you said that you will describe any sharing of tasks with people not formally qualified to provide care. How do you know they are not formally qualified; is it non-hospital staff (as family members etc could be doctors or nurses) or will you include hospital staff who are undertaking tasks they are formal not qualified to do so?

Within the Kenyan health system there are group of cadres allowed to provide care with formal training and regulation. However, in some settings, based on anecdotal knowledge, you might find other persons not formally trained or regulated providing care due to nurse shortages for instance support staff (ideally meant for non-clinical/nursing tasks) or mothers undertaking tasks considered by nurses themselves as tasks that should currently only be performed by nurses according to Kenyan recommendations. In this study we are will not be focusing on health care workers undertaking tasks they are not formally qualified for but are interested in essential nursing task being provided by non-healthcare providers (informal task-sharing).

Ethics

1) Does the ethical approval provided for the study cover all hospital sites that you are visiting?

Need to state here if it does and if it doesn’t that you are applying for ethical approval where needed.

Ethical approval was provided for the study to cover all hospitals. However, permission to conduct the study was sought from each hospital’s administration.

“Ethical approval to conduct this study in all hospitals has been granted by the Kenya Medical Research Institute (KEMRI) Scientific and Ethics Review Unit (Approval No. KEMRI/SERU/CGMR-C/065/3404) while permission to undertake the study in the respective hospitals was sought from each of the hospital’s administrative offices”

Strengths and Limitations

1) (This links in with my first point under general comments). You appear to already know the answer to your question. In the last point you state that this study will be highlighting human resource issues that warrant attention and will influence recommendations on staffing norms. Missed care could be the result of human resources issues but the concern could be with the availability of staff, with the required training, and not staffing recommendations. And if it is due to the availability of staff then your recommendations will need to take account of the nursing staff that are available to work in newborn units.

We appreciate the concern raised. To clarify and as highlighted in the protocol, this study work is part of a broader programme of work that has highlighted gaps in human resources. However, the implications of these shortages on nursing care provided have not been quantified/documentated. As part of this study we are collecting additional data on nurse staffing and workload (number of newborns in the unit) during the observation period.

“Additional data to be collected related to each block of observation time will include data on the nurse patient ratio for the ward as a whole, what additional staff are present within the unit e.g. nursing students, support staff etc. and patient workloads within the different sections of the ward.”

Introduction

1) Page 3/17, lines 40-41: “As a consequence about 45% of child mortality is neonatal mortality[1]” I would state that the 45% of child mortality is for children under 5 years of age.

This has now been rephrased.

“As a consequence, about 45% of mortality for children under-5 years is attributable to neonatal mortality”

Methods and analysis

1) Page 4/17, lines 55: Need to clarify if the two health facilities that have declined to participate in other studies are being excluded and how this exclusion could impact on your findings.

The two facilities that declined in the broader study were not considered for this study. The characteristics of these hospitals were not different and were therefore not likely to introduce selection bias. Furthermore, the selection of health facilities was purposive, as support by the hospital administration was important due to the nature of data collection and the sensitivity associated with direct observation of care.

2) Reference where you are getting your figures from.

- 34 facilities (100% of 24/7 care)
- 3 excluded (so these three units combined provide 1% of 24/7 care?)
- 31 facilities (99% of 24/7 care)
- 13 facilities with an admission rate of >100 newborns (96% of 24/7 care)
- So that means that 18 facilities, with 24/7 care, account for 3% (99%-96%) of admissions? And for the 13 selected facilities that provide ~7.4% of 24/7 care each?

As outlined in the manuscript, the current study is part of wider programme of work aiming to understand health service delivery of newborns, as such, we are building on findings and lessons from the completed phases of the wider study. We have revised the manuscript text and referenced findings from earlier work. The referenced manuscripts have now been accepted for publication and will be available by the time this manuscript is published.

“The proposed research work will be undertaken as part of a broader set of work being conducted in collaboration with Nairobi City County. This collaboration includes work to characterise all the facilities providing inpatient newborn care 24 hours, 7 days a week (hereafter referred to as 24/7) in Nairobi [26] and ethnographic work to understand the wider context and practice of neonatal nursing. Based on findings from the broader study, Nairobi

county has 34 health facilities providing 24/7 inpatient newborn care, of these 2 health facilities declined to take part in prior quality of care surveys[27]. Excluding the military health facility with restrictive admission policy, the remaining 31 health facilities that form the population for this study provide 99% of all inpatient neonatal care.”

3) Need more information on how and why the six centres were ‘purposefully selected’. Need to ensure that selection bias is not taking place.

Facilities that met the inclusion criteria were stratified by sector and workload. Furthermore, the selection of health facilities was purposive, as support by the hospital administration was important due to the nature of data collection and the sensitivity associated with direct observation of care. As such, we cannot rule out selection bias as facilities providing better care might have been more likely to agree to partake in the study. However, this is the first such study using direct observational methods to quantify missed care and in a LMIC and therefore the findings will still be important.

4) Page 6/17, lines 25-32: Step 3. Without doing a pilot how to you know this is feasible and what is your back up plan?

Thank you for this comment. As part of the study, we had a 6 weeks pilot phase where the feasibility of the study and testing of data collections was done. We have expounded the section piloting for clarity.

“Prior to the start of the study, the observation checklist will be extensively piloted over a period of 6 weeks to determine the quantity and quality of data that can be reasonably gathered and will be adapted as needed. The tool will be piloted in one public health facility that will not be used as a study site in the final study by the research assistant who will subsequently be responsible for training the data clerks. During piloting, we will observe care provision in each of the nursing 12-hour time blocks to explore what tasks can be observed, what number of newborns will be logistically feasible to observe, the different nursing routines in the different shifts and the documents used for reporting on nursing activities.”

5) For sample sizes and estimates I have requested that a statistician reviews this section, as I have concerns about the sampling but would value the opinion of a statistician. Also, you have not mentioned that you have received any statistically support/ advice for this protocol and I would advise seeking their guidance.

We would like to confirm that we sought statistical advice for this protocol.

6) Page 7/17, Procedures:

- You are extracting medical information at the start of the shift – when is parental consent being sought?

We have responded to a similar comment above and improved on the text for clarity. In brief, the start of the observational periods are preceded by the 3-hourly feeding sessions (6am or 6pm) before the 7am time-point when we start the observations. This provides adequate time to get consent. If a mother indicates they need more time to decide or consult, this is allowed for and another mother is considered for inclusion into the study. Findings from our pilot work suggested that extracting data from the medical records would take about 5 minutes for each baby as the details required were limited. We have revised the text to improve on clarity on the consenting process.

“Written informed consent will be sought from the mothers of babies considered for direct observations at the start of each observation period. In Kenya, pregnant adolescents between ages (15 – 17 years) are considered ‘emancipated minors’ and their written informed consent was obtained. The start of the direct observation shift (7am or 7pm) is just before when mothers are in the newborn unit for the 3-hourly feeding session at 6am or 6pm, as such mothers will be approached during this period. In instances, where mothers indicate more time is required to make a decision or to consult they will be allowed to do so and another mother whose baby meets the criteria for observation will be approached”

- Determining the total number of tasks at the start of the shift could be time consuming for non-nursing observers. What happens if the observer misses tasks as they are going through the notes?

Estimating the number task at the beginning of the shift involves checking the disease severity (category A, B, C) and if the baby is receiving a specialized intervention like phototherapy,

oxygen, IV fluids etc. Our experience during the pilot phase indicated that this often took about 5 minutes or less per baby as the range of interventions available is very restricted in LMIC settings as ours. It was also unlikely that the observer would miss the task as most of them involves an equipment being attached to the baby.

- Lines 27: you state the 12 hour shift under observation is from 7am to 7pm but if you are including nights than shouldn't this be 7am-7pm or 7pm-7am?

We have made this change in the manuscript.

“For each of the newborns selected to participate in the study, we will make direct observations on how often certain routine nursing tasks (listed in table 1) are undertaken in a 12-hour shift (7am -7pm or 7pm to 7am) using an observation checklist.”

- Routine and critical tasks can differ based on day or night shift how are you accounting for this?

We acknowledge that care might differ by day or night shift, we anticipated that some tasks might be done in the day only or the night only and this might differ from hospital to hospital. By randomizing and stratifying observations by day/night, it allows us to provide a general estimate (of combined night and day shifts) that takes account of possible variation.

- Revising expected tasks if a child changes area also takes time and this will mean that the observer will not be able to collect data. It could also mean depending on where the child is transfer to within the unit that the observer will not be able to collect data any more on that child.

Thank you for this comment. This was considered carefully and in our opinion, this was the only feasible way of to tackle the challenge of babies moving/having changing needs. It is important to recognize that if a baby moves then the denominator of observable tasks also reflects this.

- How will you handle incomplete data – if the observer can no longer collect data on a child due to being discharged, transferred out of unit or to another area within the hospital? How does this impact on estimates and sample sizes etc?

The sample size presented is for illustrative purposes and based on the level of precision we can report and is not estimated to test a hypothesis. The aim of the study is to describe of the essential neonatal care required, to what degree is this care missed. We acknowledge and have explained in the manuscript that there are instances where data may not be collected if a baby is discharged or changes condition. These data will be truncated at this point and missing data excluded from the denominator.

“Observations will be stopped if a baby is transferred out of a section or discharged, however, we will use the data collected up to the point of exit and the number of observation hours will be documented. If the babies’ condition changes and their category changes but they remain in the same observation area, we will document this change and revise the expected number of tasks. In both instances the effective denominator for expected nursing tasks will be changed as required.”

- Page 8/17, lines 13-16. You discuss piloting the study – I would include a lot more information on this, as how the pilot is done will affect how the study is run.

We have now revised the manuscript to explain how the piloting phase has been used in to inform the study.

“Prior to the start of the study, the observation checklist was extensively piloted over a period of 6 weeks to determine the quantity and quality of data that can be reasonably gathered and was adapted as needed. The tool was piloted in one of the public health facility by the research assistant who was also responsible for training the data clerks. During piloting, we observed care provision in each of the nursing 12-hour time blocks to explore what tasks would be observed, what number of number newborns would be logistically feasible to observe, the different nursing routines in the different shifts and the documents where the various nursing tasks were documented.”

- Table 1: how frequently are you expecting each of the routine and critical tasks to be done, at a minimum?

We have presented in the observation checklist, now included in the manuscript, the minimum number of tasks we expect.

References

1) Not all references are typed out correctly, for example:

Page 13/17, lines: 5: "1. WHO U. A Decade of Tracking Progress for Maternal, Newborn and Child Survival. 2015."

References have now been revised and updated.

Reviewer: 3

Reviewer Name: John Harris

Institution and Country: University of Pittsburgh, USA

Competing Interests: None declared.

The authors present a study protocol for a cross-sectional study of nurses using structured observations of newborn nursing care to quantify the amount of "missed care" (appropriate nursing care not provided) in a lower-middle income country.

The protocol does a good job of establishing the need to improve newborn care in low and middle income country healthcare. Overall, the description of the study is thorough and detailed to adequately inform a reader. The description of sampling methodology and sample size calculations are especially clear.

Comments:

1. Further description of why the study method was chosen would be helpful. Structured observations is a reasonable methodology, but it would be helpful to contrast it with other methods including surveys, simulated patient exercises, video observation, semi-structure interviews, focus groups, etc.

Thank you for this comment. We have highlighted as part of our discussion that other methods were considered. Previous studies have used interviews, focus group discussions and self-reported nurse surveys. However, limitations to these methods have been linked to reporting bias where nurses may not explicitly report all missed care as a way of protecting their professional allegiances and values. An optimal approach to documenting missed care would be use of video recording but based on informal discussions with stakeholders it was deemed not feasible at this time due to the ethical, medico-legal and administrative challenges.

We have now revised the text in the discussion to highlight this.

"In fact, a recent review by Jones and colleagues identified that only questionnaires have previously been used to quantify missed care and these might be limited by reporting bias [22]. However, we acknowledge that direct observational methods have limitations on the number and actual tasks that can be observed, might be influenced by observer bias and are at risk of Hawthorne effect. As such, the tasks in our observation checklist are limited to essential neonatal nursing tasks that can be observed while those linked to documentation will involve review of medical records at the nurses' desk since documentation of care is done at the nurses' desk and not at the cot-side. An alternative method for data collection that we considered was the use of video in the newborn unit with later evaluation of what is done (or not done). However, informal discussions indicated this might be controversial at this stage due to the ethical and medical legal issues that might emanate from this approach and the administrative approvals required for video recording in hospitals, nonetheless this is a potential area for future research."

2. "Missed care" has at least two etiologies: process inconsistency (care can't be provided because of interrupting tasks, inadequate staffing, etc) and inadequate knowledge or training (so care is not provided because the provider doesn't realize how to do it properly). The present methodology will group both problems into one episode of "missed care". The methodology doesn't have an obvious way to differentiate between these two problems.

Thank you for this comment. We acknowledge that missed care might result from process inconsistency or inadequate knowledge. The focus of this study is to evaluate missed care from a process inconsistency perspective since the tasks we are evaluating are basic routine neonatal

nursing tasks that are unlikely to be missed due to knowledge gap. We have now included the observation checklist (supplementary file 1) to illustrate the tasks that were of interest.

3. Beyond collecting information about the baby's condition and the checklist, is there any other information being collected? Qualitative observations noting the chaos, organization, methods the nurse used to complete tasks, experience of the nurse, training/background of the nurse, how the nurse interacted with other staff, etc would be very helpful.

Thank you for this suggestion. Although, this study is mainly quantitative, we will collect additional data as field notes in the observational checklist comments section to describe additional activities that might have influenced the delivery of care. For instance, if there was an ongoing resuscitation for an extended period of time. In addition, we are collecting workload and staffing data for the observation periods. As part of the broader project, there is ongoing ethnographic work to describe these organizational and chaotic processes that go on within the newborn units. Data from this observational study will be complemented by ethnographic work to better describe nursing care provision within newborn units

4. An exact checklist was not provided, but hopefully it goes into enough detail to give some sense of whether a complex task like "Communication/counselling parents" is done in an appropriate or inappropriate manner. More information about how the checklist was created would be of interest to the reader.

We have now included the observation checklist as a supplementary file. The aim of this study is not to describe how well a task is done but to quantify the degree to which essential neonatal nursing tasks get done at all. To achieve the former, we would require a qualified nurse or health care provider who understands the details of the various tasks but this was limited by the ethical, medical legal and administrative challenges of having a practitioner who might be required to intervene during emergencies in hospitals where they are not allowed to practice.

VERSION 2 – REVIEW

REVIEWER	John Harris University of Pittsburgh, USA
REVIEW RETURNED	15-May-2018

GENERAL COMMENTS	The responses to comments are appropriate and improve the protocol considerably. I recommend accepting this protocol for publication.
---

REVIEWER	Hazel A Smith Trinity College Dublin, Ireland and Our Lady's Children's Hospital, Crumlin, Ireland
REVIEW RETURNED	17-May-2018

GENERAL COMMENTS	Thank you for giving me the opportunity to review your manuscript. My notes are: 1) Where possible I would avoid using the same word in a sentence. For example, I would change 'However, due to high patient workloads and limited resources, nurses may often consciously or unconsciously have to prioritise the tasks and the care they provide resulting in some tasks' to 'However, due to the high patient workloads and limited resources, nurses may often consciously or unconsciously prioritise the care they provide resulting in some tasks being.....' 2) If possible expand on the last sentence in the first paragraph in the abstract (Introduction) by stating where ('missed care, examined by direct observation, has not previously been the subject of research'). Are you referring to low income countries or everywhere? 3) For figures less than 10 I would write the word out instead of
--

	giving the number. 4) You have used the future tense but according to the dates this study is near completion. Please correct where needed. 5) Need to include 'is' in the first sentence of 'Strengths and limitations' so it reads 'The use of direct observational methods to quantifying nursing care delivered or left undone is an approach that have has not been previously used in LMICs.' 6) For the two health facilities that refused to take part I would give a sentence or two on these two centres and state any concerns you may have with selection bias. It could be that they would not meet your inclusion criteria anyway – but if this is the case I would state it. 7) In methods and analysis - study populations: Need to state what happens if you start to collect data on a child who later becomes critically ill? 8) Also, need to state how you plan on dealing with any missing or incomplete data 9) For sample size, on page, need to give references to justify your statements. For example 'For tasks that would be conducted with an expected frequency of once in 12 hours (eg. IVintravenous pencillin administration) we might therefore observe 216 task opportunities. Taking a (statistically) conservative assumption that on 50% of expected occasions the task is observed to be done then we could report this with a precision of $\pm 13.4\%$'. Tell the reader where the 13.4% come from. 10) On page 8 the word 'undertaking' is misspelt in heading 'Practical considerations to for undertakinge direct observations' 11) On page 10 you state 'It will be made clear that at any stage nurses or parents can withdraw consent/permission for observation, temporarily or for the rest of a shift, without explanation and with no penalty' but are you not introducing a bias by allowing periods of the observation period to be excluded from data collection? 12) On your questionnaire, 'Quantifying neonatal nursing care observational checklist', section bio data, question 10 states 'Age (days/ If day 1 of life give hours)'. But as you are not collecting date of birth how will you distinguish between a '2' that refers to hours and a '2' that refers to days? 13) Check your referencing software. For example, for references 15 & 16 the year of publication is missing.
--	--

VERSION 2 – AUTHOR RESPONSE

Response to reviewer comments.

1) Where possible I would avoid using the same word in a sentence. For example, I would change 'However, due to high patient workloads and limited resources, nurses may often consciously or unconsciously have to prioritise the tasks and the care they provide resulting in some tasks' to 'However, due to the high patient workloads and limited resources, nurses may often consciously or unconsciously prioritise the care they provide resulting in some tasks being.....'

Thank you for this suggestion. We have now revised the text as suggested.

“However, due to high patient workloads and limited resources, nurses may often consciously or unconsciously prioritise the care they provide resulting in some tasks being left undone or partially done (missed care)”

2) If possible expand on the last sentence in the first paragraph in the abstract (Introduction) by stating where ('missed care, examined by direct observation, has not previously been the subject of research'). Are you referring to low income countries or everywhere?

Thank you for this comment. We have now revised the text for clarity.
“However, missed care, examined by direct observation, has not previously been the subject of research in low income countries”

3) For figures less than 10 I would write the word out instead of giving the number.

We have now revised the text as per above suggestion.

4) You have used the future tense but according to the dates this study is near completion. Please correct where needed.

Thank you for this comment. We are not very clear about this comment as the manuscript was submitted while the study was undergoing and future tense is the recommended language for protocols.

5) Need to include 'is' in the first sentence of 'Strengths and limitations' so it reads 'The use of direct observational methods to quantifying nursing care delivered or left undone is an approach that have has not been previously used in LMICs.'

Thank you for this comment. We have now revised text as suggested.
“The use of direct observational methods to quantifying nursing care delivered or left undone is an approach that has not been previously used in LMICs”

6) For the two health facilities that refused to take part I would give a sentence or two on these two centres and state any concerns you may have with selection bias. It could be that they would not meet your inclusion criteria anyway – but if this is the case I would state it.

Thank you for this comment. We have now included a statement describing the 2 facilities that declined to partake the study and an additional statement to clarify that this would not introduce selection bias.

Study site, 1st paragraph

“Based on findings from the broader study, Nairobi county has 34 health facilities providing 24/7 inpatient newborn care, of these, two small health facilities declined to take part in prior quality of care surveys and were estimated to have less than 50 neonatal admissions each per year”

Stud site, 2nd paragraph

“.....As such, the two health facilities that decline to participate in prior quality of care surveys do not meet this criterion and are unlikely to introduce selection bias”

7) In methods and analysis - study populations: Need to state what happens if you start to collect data on a child who later becomes critically ill?

Thank you for this comment. We have expanded the text in the first paragraph of the 'procedures' section to clarify that we stopped observations for children who were became critically sick not meet the criteria for the minimum nursing standards. Otherwise we have already explained that children who changed condition and were transferred to another section where observations would still be undertaken would still remain under observation.
“Observations will be stopped if a baby is transferred out of a section, changes condition and becomes critically ill (requires specialised treatment which the minimum draft standards do not apply) or discharged, however, we will use the data collected up to the point of exit and the number of observation hours will be documented.”

“If the babies' condition changes and their category changes but they remain in the same observation area, we will document this change and revise the expected number of tasks. In both instances the effective denominator for expected nursing tasks will be changed as required.”

8) Also, need to state how you plan on dealing with any missing or incomplete data.

Thank you for this comment. We have now included a statement in the methods section on how missing data will be handled.

“Missing or incomplete data will be coded as a category and where necessary presented as such. When reporting on effective number of tasks done (or left undone), missing/incomplete data will be excluded from the effective denominator the task would have contributed to and hence avoid spurious inflation of the denominator”

9) For sample size, on page, need to give references to justify your statements. For example 'For tasks that would be conducted with an expected frequency of once in 12 hours (eg. IVintravenous pencillin administration) we might therefore observe 216 task opportunities. Taking a (statistically) conservative assumption that on 50% of expected occasions the task is observed to be done then we could report this with a precision of $\pm 13.4\%$ '. Tell the reader where the 13.4% come from.

Thank you for this comment. Our sample size calculation is a precision based approach that seeks to illustrate within what precision levels we would report our estimates given the number of observed tasks. we have explained this in the first paragraph of the sample size section (see below).

“This is an exploratory cross-sectional study and as such, we illustrate the precision with which we can report proportions of tasks done (or not done) assuming different sized denominators. The size of the denominator is related to the recommended frequency that tasks should be performed (see figure 2). To estimate the precision of reporting we have used a sample estimation approach for cluster designs and assuming a design effect of 2 to adjust for clustering of observed tasks around individual newborns within hospitals.”

We have also reworded the above sentence for clarity.

“For tasks that would be conducted with an expected frequency of once in 12 hours (e.g. intravenous penicillin administration) we might therefore observe 216 task opportunities. Taking a (statistically) conservative assumption that on 50% of expected occasions the task is observed to be done then we could report the proportion of such a task being done with a precision of $\pm 13.4\%$.”

10) On page 8 the word 'undertaking' is misspelt in heading 'Practical considerations to for undertaking direct observations'

This typographical error is now corrected.

11) On page 10 you state 'It will be made clear that at any stage nurses or parents can withdraw consent/permission for observation, temporarily or for the rest of a shift, without explanation and with no penalty' but are you not introducing a bias by allowing periods of the observation period to be excluded from data collection?

Thank you for this comment. We acknowledge, that there is a risk of introducing bias by allowing periods of observation to be excluded from data collection. Where such instances occur, we will document and report for what proportion of our observations this was the case. However, we anticipated that there might be instances where nurses or parents might want privacy may be during communication with the health care providers but is okay with direct observation of care. We have also previously explained that in instances where observations are interrupted, we will revise the number of expected tasks and effective denominators accordingly.

12) On your questionnaire, 'Quantifying neonatal nursing care observational checklist', section bio data, question 10 states 'Age (days/ If day 1 of life give hours)'. But as you are not collecting date of birth how will you distinguish between a '2' that refers to hours and a '2' that refers to days?

Thank you for this comment. In the standard operating procedures for data collection, it is outlined that where hours are documented, the units should be indicated.

13) Check your referencing software. For example, for references 15 & 16 the year of publication is missing.

The references have been corrected and the year included

VERSION 3 – REVIEW

REVIEWER	Hazel A Smith Trinity College Dublin, Ireland and Our Lady's Children's Hospital, Crumlin, Ireland
REVIEW RETURNED	17-May-2018

GENERAL COMMENTS	Thank you for giving me the opportunity to review your manuscript. My notes are:  1) Where possible I would avoid using the same word in a sentence. For example, I would change 'However, due to high patient workloads and limited resources, nurses may often consciously or unconsciously have to prioritise the tasks and the care they provide resulting in some tasks' to 'However, due to the high patient workloads and limited resources, nurses may often consciously or unconsciously prioritise the care they provide resulting in some tasks being.....' 2) If possible expand on the last sentence in the first paragraph in the abstract (Introduction) by stating where ('missed care, examined by direct observation, has not previously been the subject of research'). Are you referring to low income countries or everywhere? 3) For figures less than 10 I would write the word out instead of giving the number. 4) You have used the future tense but according to the dates this study is near completion. Please correct where needed. 5) Need to include 'is' in the first sentence of 'Strengths and limitations' so it reads 'The use of direct observational methods to quantifying nursing care delivered or left undone is an approach that have has not been previously used in LMICs.' 6) For the two health facilities that refused to take part I would give a sentence or two on these two centres and state any concerns you may have with selection bias. It could be that they would not meet your inclusion criteria anyway – but if this is the case I would state it. 7) In methods and analysis - study populations: Need to state what happens if you start to collect data on a child who later becomes critically ill? 8) Also, need to state how you plan on dealing with any missing or incomplete data 9) For sample size, on page, need to give references to justify your statements. For example 'For tasks that would be conducted with an expected frequency of once in 12 hours (eg. IVintravenous pencillin administration) we might therefore observe 216 task opportunities. Taking a (statistically) conservative assumption that on 50% of expected occasions the task is observed to be done then we could report this with a precision of $\pm 13.4\%$'. Tell the reader where the 13.4% come from. 10) On page 8 the word 'undertaking' is misspelt in heading 'Practical considerations to for undertaking direct observations' 11) On page 10 you state 'It will be made clear that at any stage nurses or parents can withdraw consent/permission for observation, temporarily or for the rest of a shift, without explanation and with no penalty' but are you not introducing a bias by allowing periods of the observation period to be excluded from data collection? 12) On your questionnaire, 'Quantifying neonatal nursing care observational checklist', section bio data, question 10 states 'Age (days/ If day 1 of life give hours)'. But as you are not collecting date of birth how will you distinguish between a '2' that refers to hours and a '2' that refers to days? 13) Check your referencing software. For example, for references 15 & 16 the year of publication is missing.
--

REVIEWER	John Harris University of Pittsburgh, USA
REVIEW RETURNED	15-May-2018

GENERAL COMMENTS	The responses to comments are appropriate and improve the protocol considerably. I recommend accepting this protocol for publication.
---

VERSION 3 – AUTHOR RESPONSE

Response to reviewer comments.

1) Where possible I would avoid using the same word in a sentence. For example, I would change 'However, due to high patient workloads and limited resources, nurses may often consciously or unconsciously have to prioritise the tasks and the care they provide resulting in some tasks' to 'However, due to the high patient workloads and limited resources, nurses may often consciously or unconsciously prioritise the care they provide resulting in some tasks being.....'

Thank you for this suggestion. We have now revised the text as suggested.

“However, due to high patient workloads and limited resources, nurses may often consciously or unconsciously prioritise the care they provide resulting in some tasks being left undone or partially done (missed care)”

2) If possible expand on the last sentence in the first paragraph in the abstract (Introduction) by stating where ('missed care, examined by direct observation, has not previously been the subject of research'). Are you referring to low income countries or everywhere?

Thank you for this comment. We have now revised the text for clarity.

“However, missed care, examined by direct observation, has not previously been the subject of research in low income countries”

3) For figures less than 10 I would write the word out instead of giving the number.

We have now revised the text as per above suggestion.

4) You have used the future tense but according to the dates this study is near completion. Please correct where needed.

Thank you for this comment. We are not very clear about this comment as the manuscript was submitted while the study was undergoing and future tense is the recommended language for protocols.

5) Need to include 'is' in the first sentence of 'Strengths and limitations' so it reads 'The use of direct observational methods to quantifying nursing care delivered or left undone is an approach that have has not been previously used in LMICs.'

Thank you for this comment. We have now revised text as suggested.

“The use of direct observational methods to quantifying nursing care delivered or left undone is an approach that has not been previously used in LMICs”

6) For the two health facilities that refused to take part I would give a sentence or two on these two centres and state any concerns you may have with selection bias. It could be that they would not meet your inclusion criteria anyway – but if this is the case I would state it.

Thank you for this comment. We have now included a statement describing the 2 facilities that declined to partake the study and an additional statement to clarify that this would not introduce selection bias.

Study site, 1st paragraph

“Based on findings from the broader study, Nairobi county has 34 health facilities providing 24/7 inpatient newborn care, of these, two small health facilities declined to take part in prior quality of care surveys and were estimated to have less than 50 neonatal admissions each per year”

Stud site, 2nd paragraph

“.....As such, the two health facilities that decline to participate in prior quality of care surveys do not meet this criterion and are unlikely to introduce selection bias”

7) In methods and analysis - study populations: Need to state what happens if you start to collect data on a child who later becomes critically ill?

Thank you for this comment. We have expanded the text in the first paragraph of the 'procedures' section to clarify that we stopped observations for children who were became critically sick not meet the criteria for the minimum nursing standards. Otherwise we have already explained that children who changed condition and were transferred to another section where observations would still be undertaken would still remain under observation. *"Observations will be stopped if a baby is transferred out of a section, changes condition and becomes critically ill (requires specialised treatment which the minimum draft standards do not apply) or discharged, however, we will use the data collected up to the point of exit and the number of observation hours will be documented."*

"If the babies' condition changes and their category changes but they remain in the same observation area, we will document this change and revise the expected number of tasks. In both instances the effective denominator for expected nursing tasks will be changed as required."

8) Also, need to state how you plan on dealing with any missing or incomplete data.

Thank you for this comment. We have now included a statement in the methods section on how missing data will be handled.

"Missing or incomplete data will be coded as a category and where necessary presented as such. When reporting on effective number of tasks done (or left undone), missing/incomplete data will be excluded from the effective denominator the task would have contributed to and hence avoid spurious inflation of the denominator"

9) For sample size, on page, need to give references to justify your statements. For example 'For tasks that would be conducted with an expected frequency of once in 12 hours (eg. IVintravenous pencillin administration) we might therefore observe 216 task opportunities. Taking a (statistically) conservative assumption that on 50% of expected occasions the task is observed to be done then we could report this with a precision of $\pm 13.4\%$ '. Tell the reader where the 13.4% come from.

Thank you for this comment. Our sample size calculation is a precision based approach that seeks to illustrate within what precision levels we would report our estimates given the number of observed tasks. we have explained this in the first paragraph of the sample size section (see below).

"This is an exploratory cross-sectional study and as such, we illustrate the precision with which we can report proportions of tasks done (or not done) assuming different sized denominators. The size of the denominator is related to the recommended frequency that tasks should be performed (see figure 2). To estimate the precision of reporting we have used a sample estimation approach for cluster designs and assuming a design effect of 2 to adjust for clustering of observed tasks around individual newborns within hospitals."

We have also reworded the above sentence for clarity.

"For tasks that would be conducted with an expected frequency of once in 12 hours (e.g. intravenous penicillin administration) we might therefore observe 216 task opportunities. Taking a (statistically) conservative assumption that on 50% of expected occasions the task is observed to be done then we could report the proportion of such a task being done with a precision of $\pm 13.4\%$."

10) On page 8 the word 'undertaking' is misspelt in heading 'Practical considerations to for undertaking direct observations'

This typographical error is now corrected.

11) On page 10 you state 'It will be made clear that at any stage nurses or parents can withdraw consent/permission for observation, temporarily or for the rest of a shift, without explanation and with no penalty' but are you not introducing a bias by allowing periods of the observation period to be excluded from data collection?

Thank you for this comment. We acknowledge, that there is a risk of introducing bias by allowing periods of observation to be excluded from data collection. Where such instances occur, we will document and report for what proportion of our observations this was the case. However, we anticipated that there might be instances where nurses or parents might want privacy may be during communication with the health care providers but is okay with direct

observation of care. We have also previously explained that in instances where observations are interrupted, we will revise the number of expected tasks and effective denominators accordingly.

12) On your questionnaire, 'Quantifying neonatal nursing care observational checklist', section bio data, question 10 states 'Age (days/ If day 1 of life give hours)'. But as you are not collecting date of birth how will you distinguish between a '2' that refers to hours and a '2' that refers to days?

Thank you for this comment. In the standard operating procedures for data collection, it is outlined that where hours are documented, the units should be indicated.

13) Check your referencing software. For example, for references 15 & 16 the year of publication is missing.

The references have been corrected and the year included